# Sleeping duration, physical activity, alcohol drinking and other risk factors as potential attributes of metabolic syndrome in adults in Ethiopia: A hospital-based cross-sectional study

Mulugeta Belayneh[1], Tefera Chane Mekonnen[ID][2]*, Sisay Eshete Tadesse[2], Erkihun Tadesse Amsalu[3], Fentaw Tadese[3]

**1** Public Health Specialist at Dessie Comprehensive Specialized Hospital, Dessie, North Eastern Ethiopia, **2** Academician and Researcher at School of Public Health, College of Medicine and Health Sciences, Wollo University, Dessie, Ethiopia, **3** Department of Epidemiology and Biostatistics, School of Public Health, College of Medicine and Health Sciences, Wollo University, Dessie, Ethiopia

\* teferachane@gmail.com

## Abstract

### Background

Available evidence showed that metabolic syndrome in the adult population is persistently elevated due to nutrition transition, genetic predisposition, individual-related lifestyle factors, and other environmental risks. However, in developing nations, the burden and scientific evidence on the pattern, and risk exposures for the development of the metabolic syndrome were not adequately investigated. Thus, the study aimed to measure the prevalence of metabolic syndrome and to identify specific risk factors among adult populations who visited Dessie Comprehensive Specialized Hospital, Ethiopia.

### Methods

A hospital-based cross-sectional study was conducted among randomly selected 419 adults attending Dessie Comprehensive Specialized Hospital from January 25 to February 29, 2020. We used the WHO STEP-wise approach for non-communicable disease surveillance to assess participants' disease condition. Metabolic syndrome was measured using the harmonized criteria recommended by the International Diabetes Federation Task Force in 2009. Data were explored for missing values, outliers and multicollinearity before presenting the summary statistics and regression results. Multivariable logistic regression was used to disentangle statistically significant predictors of metabolic syndrome expressed using an odds ratio with a 95% of uncertainty interval. All statistical tests were managed using SPSS version 26. A non-linear dose-response analysis was performed to show the relationships between metabolic syndromes with potential risk factors.

### Results

The overall prevalence of metabolic syndrome among adults was 35.0% (95% CI, (30.5, 39.8)). Women were more affected than men (i.e. 40.3% vs 29.4%). After adjusting for other variables, being female [OR = 1.85; 95% CI (1.01, 3.38)], urban residence [OR = 1.94; 95%

**Data Availability Statement:** All relevant data can be accessed at Dryad repository https://datadryad.org/stash/share/

7LdGhNH656AkYgUjnqR5S3v82TdUdD-svZisGVhTx8A.

**Funding:** Wollo University and Dessie Comprehensive Specialized Hospital provided a monthly salary for the Authors as clearly specified in the manuscript. However, the funders had no role in study design, data collection and analysis, decision to publish, or preparation of the manuscript.

**Competing interests:** The authors have declared that no competing interests exist.

**Abbreviations:** AOR, Adjusted Odds Ratio; BMI, Body-Mass Index; BP, Blood Pressure; COR, Crude Odds Ratio; DCSH, Dessie Comprehensive Specialized Hospital; FBS, Fasting Blood Sugar; HDL, High-Density Lipoprotein; MetS, Metabolic Syndrome; NCD, Non-Communicable Disease; OPD, Outpatient Department; WC, Waist Circumference.

CI (1.08, 3.24)], increased age [OR = 18.23; 95% CI (6.66, 49.84)], shorter sleeping durations [OR = 4.62; 95% CI (1.02, 20.98)], sedentary behaviour [OR = 4.05; 95% CI (1.80, 9.11)], obesity [OR = 3.14; 95% CI (1.20, 8.18)] and alcohol drinking [OR = 2.85; 95% CI (1.27,6.39)] were positively associated with the adult metabolic syndrome. Whilst have no formal education [OR = 0.30; 95% CI (0.12, 0.74)] was negatively associated with metabolic syndrome.

## Conclusions

The prevalence of adult metabolic syndrome is found to be high. Metabolic syndrome has linear relationships with BMI, physical activity, sleep duration, and level of education. The demographic and behavioural factors are strongly related with the risk of metabolic syndrome. Since most of the factors are modifiable, there should be urgent large-scale community intervention programs focusing on increased physical activity, healthy sleep, weight management, minimize behavioural risk factors, and healthier food interventions targeting a lifecycle approach. The existing policy should be evaluated whether due attention has given to prevention strategies of NCDs.

## Introduction

Metabolic syndrome (MetS) is defined by a constellation of interconnected factors that directly increases the risk of cardiovascular disease (CVD), type 2 diabetes mellitus, and all-cause mortality [1]. Among other etiologies of MetS insulin resistance and visceral adiposity are highly responsible for chronic inflammation process characterized by the production of abnormal adipocytokines including tumor necrosis factor α, interleukin-1 (IL-1), IL-6, leptin, and adiponectin. The interaction between components of the clinical phenotype of the syndrome with its biological phenotype contributes to the development of a pro-inflammatory state and further a chronic, subclinical vascular inflammation which modulates and results in atherosclerotic processes [2–5].

MetS is present if three or more of the following five criteria are met: waist circumference (WC) > 83.7 cm for males and ≥ 78.0 cm for females [6]; fasting blood glucose(FBG) ≥100 mg/dL (5.5 mmol/L) or treatment with hypoglycaemic agents or insulin; systolic blood pressure ≥130 or diastolic blood pressure ≥85 mm Hg or antihypertensive drug treatment; serum triglycerides ≥150 mg/dL (1.7 mmol/L) or treatment for hypertriglyceridemia and high-density lipoprotein (HDL-C)<40 mg/dL (1.0 mmol/L) for men or <50 mg/dL (1.3 mmol/L) in women [7].

However, in 2009 the International Diabetic Federation (IDF) Task Force revised above criteria to diagnosis MetS by including central obesity (defined when WC >94 cm for men and >80 cm for women) plus any two of the remaining four criteria listed above. The current study used this harmonized definition made to assess MetS [7–11, 25].

Against this backdrop, MetS is being considered a rising public health issue globally, which ranges from 10% to 50% [6, 10]. While in Africa the prevalence ranged from 17% to 25% [12]. Ethiopia ranked among the top four countries of 15 Eastern Sub-Saharan Africa countries in terms of mortality and disability-adjusted life-years based on the age-standardized proportion of disease attributable to dietary and metabolic risks.

Adults with MetS are twice as likely to die and three times as likely to have a heart attack or stroke compared with people without MetS; and different articles suggest that three up to five folds greater risk of developing type II diabetes [13–17].

MetS has an association with sudden cardiac death [7] and it is not only increasing the risk of developing non-communicable diseases (NCDs); but also increases the cost of treatment for NCDs. It has been estimated that the economic burden of hypertension and other NCDs increases from 59% to 179% by 2020 [18]. NCDs are increasingly affecting low- and middle-income countries [19]. Previous studies conducted in Ethiopia have documented a high prevalence of MetS [14, 20].

Several recent reports show that consumption of atherogenic diet, sedentary lifestyle, and tobacco consumption, physical inactivity, aging, and hormonal imbalance are considered as potential risk factors for MetS [21, 22]. Recently, this syndrome has also been noted to be associated with a state of chronic, low-grade inflammation [22]. Lifestyle interventions are recommended as the initial therapies for the treatment of MetS [22]. In Ethiopia, evidence on the magnitude and risks of exposures were not inclusive and are very limited. The current study differed from former studies in use of diagnosis criteria for MetS (both of them used the National Cholesterol education program III) [14, 20], variations in target population [21] and they didn't address lifestyle factors inclusively. The way they measured physical activity and other composite variables are not clear. However, we tried to additionally assess dietary habits, sleep duration, and substance use that may lead to MetS. The present study aimed to address the prevalence of MetS and associated factors among adults in DCSH, Ethiopia.

## Methods and materials

### Study setting and design

The study employed a hospital-based cross-sectional study at Dessie comprehensive specialized hospital (DCSH) from January 25 to February 2020. The hospital is 401 km far away from the capital city, Addis Ababa to the northeast direction, located at the Center of Dessie city and is one of the frontline government hospitals in Ethiopia. It provides diversified referral services (but not limited comprehensive obstetric, general surgery, orthopaedics, chemotherapy, neurological internal medicine and psychiatric services) for more than 7 million populations from eastern Amhara and Afar regions. It is staffed with more than 800 healthcare and administrative workers. More than 300,000 patients visit the hospital annually. Adults whose ages greater than eighteen years and attending outpatient departments (OPDs) in DCSH were the target population. All adult patients attending the OPDs at DCSH after eight hours of fasting were included in the study but all pregnant mothers were excluded from the study.

We determined the total samples of 419 adults to be included in the study by considering the prevalence of MetS in Gondar, Ethiopia [13] as 45.3%, and 10% of non-response rate. Patients who were identified as fasting for the last eight hours were marked by the red colour on their card from triage. Sampling frame was prepared using identity card number for subjects whose card was labelled with red colour on daily basis until reaching the required sample. We applied a simple random sampling technique to catch-up on study subjects. All of the methods were performed in accordance with the guideline of STROBE checklist.

### Data measurement

Participants were interviewed in face-to-face manner using the modified WHO Stepwise approach for the surveillance of NCDs structured questionnaire [23]. This approach is designed to explicitly assess the risk factors of NCDs in the scope of socio-demographic, economic, medical history, biochemical, physical measurements, and lifestyle factors including dietary habits, physical activity, and substance use. Anthropometric measurements and blood sample collection were taken by six nurses and one laboratory technologist after training was given.

Blood pressure (BP) was measured by using a digital measuring device with sitting, after resting for at least 15 minutes. The measurements were taken on the left arm after removing or rolling up clothing with the palm facing upward using appropriate cuff size, with the position the cuff above the elbow and lower band is positioned 1–2 cm above the elbow joint. BP measurements were taken times with at least 3minute intervals between consecutive measurements. The mean systolic and diastolic BP from the second and third recording was analysed and documented as elevated BP when greater or equal to 130/85 mm Hg [24].

Weight and height were measured using the electronic weighing and height scales with regular monitoring and adjustment of the beam-balance. Height and weight were recorded to the nearest 0.1 cm and 0.1 kg respectively [24].

Body Mass Index (BMI) was categorized using optimal cut-off for obesity validated in Ethiopia that classifies adults as underweight/thin when the respondent's BMI $\leq$ 18.3 for males and <21.9 for females, as normal if the BMI was between 18.3–21.5 kg/m2 for males and 21.9–23.0 kg/m2 for females, as overweight if the BMI lied between 21.6–22.2 kg/m2 for males and 23.1–24.5 kg/m2 for females and as obese if the BMI was $\geq$22.2 k/m2 for males and >24.5 kg/m2 for females [6].

For measuring participants' waist circumference (WC), we used a simple flexible steel metric tape. According to WHO, central obesity was defined when WC for men and women was greater than 94 cm and 80cm respectively [25].

Blood sample collections were taken from the study subjects who attended clinical follow-up in DCSH by the laboratory technician. Five mL of blood specimen was collected from every participant to analyse participants' fasting blood sugar (FBS) and lipid profiles in the clinical chemistry laboratory using DIRUI CS-T240 automated chemistry analyzer. Triglyceride (TG) concentration was measured by standard enzymatic assays using glycerol phosphate oxidase method and defined as elevated when $\geq$ 1.7 mmol/l (150 mg/dl) for fasting samples [26]. HDL cholesterol was determined after sample pre-treatment with a precipitating reagent and centrifugation. The participants were categorized as having low HDL when it was below 40mg/dl and 50mg/dl for men and women respectively. Participants' FBS was determined using the glucose oxidase method within 30' minutes after collection of the blood samples and diagnosed as having diabetes when FBS was $\geq$100 mg/dl [23].

Familial histories of cardio-metabolic diseases from biologically related first-degree relatives were assessed through recalling of the participants and referring the physician records [27].

Physical activity was measured using the General Physical Activity Questionnaire (GPAQ) which recommended in the Ethiopian context. Physical inactivity was defined as those who had low levels of physical activity [28, 41].

The level of alcohol consumption was categorized as current alcohol users if the study participants took alcoholic drinks within 30 days preceding the study; as moderate drinkers when participants consumed standard of two drinks on a single occasion for men, one drinks on a single occasion for women and as heavy drinkers if participants consumed standard of five or more drinks on a single occasion or twenty or more drinks per week for men, four or more drinks on a single occasion or fifteen or more drinks per week for women [23]. Likewise, the participants' exposure levels for cigarette smoking were labelled as tobacco users if a person who was either a smoker or a smokeless tobacco user, or both and as smoker if someone who, at the time of the survey, smoked any tobacco product either daily or occasionally. Smokers may be further divided into two categories: i) daily smoker if someone who smoked any tobacco product at least once a day (with the exception that people who smoke every day, but not on days of religious fasting, were still classified as daily smokers) and ii) occasional smoker when someone who smoked, but not every day [29].

The dietary risk of participants for MetS was assessed using food frequency questionnaire that includes nine food groups [30]. Fruit and vegetable consumption was also assessed using questions like 'How many servings of fruit do you eat on a typical day?' and 'How many servings of vegetables do you eat on a typical day?' using 24-hour dietary recall data [28]. The WHO recommends an individual intake of at least 400g of fruits and vegetables a day, the equivalent of five servings, which was used as the cut-off for low fruit and vegetable consumption [31].

Sleep duration was assessed by the question: "In the past year, on average, how many hours/minutes of sleep (including day time naps) did you take per day?" with the following category responses: < 6 hrs, 6 to7 hrs, 8 to 9 hrs, and ≥10 hrs [32]. Although there is no common consensus on the optimal duration of sleep, some studies used the above groupings. The Center for Disease Control and Prevention advises that healthy sleep varies by sex and age. The recommended sleep duration ranges 6 hrs. to 9 hrs for adults as defined by National Sleep Foundation of America [33].

## Data analysis

After checking the completeness of the questionnaire, the data were coded and entered into Epi-Data version 4.6.0.2 then exported to SPSS version 25 for further analyses. Frequencies, percentage, mean and standard deviation were computed. Those variables with p-value less than 0.2 in the bivariable analysis were exported to the final model. Then multivariable logistic regression was performed and variables with a p-value ≤ of 0.05 were considered as significant factors and present using adjusted odds ratio (AOR), 95% CI. Model fitness was checked by Hosmer-Lemeshow test (0.970). Scatter plot was demonstrated to show the dose-response relationship (non-linear exposure variables) between MetS and physical activity, sleep duration, BMI and Educational level after logarithmic transformation of their corresponding Odds ratio.

Ethical clearance was obtained from the Research and Ethical Review Committee of the College of Medicine and Health Sciences, Wollo University. For any of the eligible study participants, the purpose, benefits, and right of withdrawal or stop filling the questionnaire were described and discussed.

## Results

### Socio-demographic characteristics

A total of 408 study subjects were involved in the study with a response rate of 97.37%. Among them, 211(51.7%) were females. The mean age of the participants was 44.74 (±15.67 of SD) years and about one-fifth of the participants, 94(23.0%), were in the age range of 18–29 years (Table 1).

### Medical and behavioural characteristics

From the total number of study participants, seventy-two (17.65%) participants had a family history of cardio-metabolic diseases, of which 33 (45.83%) of them had current MetS. Among a total of respondents, 208 (51%) of them were consuming coffee and 72 (34.6%) of them had MetS as diagnosed with the current criteria. The majority of study participants, 262(64.2%) were not involved in high level, or moderate physical activity (Table 2).

### Dietary habits and nutritional status

A higher percentage of the sample, 361 (88.5%) took insufficient fruits and vegetables. The highest number of study participants, 244(59.8%) used vegetable oil for food preparation.

**Table 1. Socio-demographic characteristics among adult patients in Dessie comprehensive specialized hospital outpatient departments Dessie, Ethiopia May 2020.**

| Variables | | Frequency | Percentage |
|---|---|---|---|
| **Sex** | Female | 211 | 51.7 |
| | Male | 197 | 48.3 |
| **Age group (in years)** | 18–29 | 94 | 23.0 |
| | 30–39 | 80 | 19.6 |
| | 40–49 | 83 | 20.3 |
| | 50–59 | 73 | 17.9 |
| | ≥60 | 78 | 19.2 |
| | Mean age (±SD) | 44.74(±15.67) | |
| **Ethnicity** | Amhara | 291 | 71.3 |
| | Oromo | 48 | 9.3 |
| | Tigrie | 38 | 11.8 |
| | Afar | 31 | 7.6 |
| **Religion** | Muslim | 193 | 47.3 |
| | Protestant | 33 | 8.1 |
| | Orthodox | 182 | 44.6 |
| **Educational status** | No formal schooling | 67 | 16.4 |
| | High school and less | 167 | 41.0 |
| | College and above | 174 | 42.6 |
| **Marital status** | Widowed | 42 | 10.3 |
| | Married | 169 | 41.4 |
| | Separated | 28 | 6.9 |
| | Divorced | 45 | 11.0 |
| | Single | 124 | 30.4 |
| **Resident** | Urban | 277 | 67.9 |
| | Rural | 131 | 32.1 |
| **Monthly Income(in ETB)** | <2000 | 79 | 19.4 |
| | 2000–4000 | 136 | 33.3 |
| | ≥4000 | 193 | 47.3 |

ETB: Ethiopian birr

From a total of study subjects, 165 (40.4%) were taken sugar and sweet daily more than one-third of them 61 (37.0%) have MetS (Table 3). Around one-fifth of the study subjects, 83 (20.34%) were overweight while 33(8.3%) of the participants were obese. The mean BMI was 23.185±3.3195 kg/m$^2$.

## Metabolic syndrome

The proportion of MetS among adults who attended DCSH was 35.0%[95% CI, (30.5, 39.5)] as measured by the 2009 harmonized definition. It was more common among women than men (40.3% vs 29.4%; p<0.023). Women had a higher percentage of reduced HDL than men (20.4% vs 11.2%; P<0.014) but no significant gender differences were observed with elevated blood pressure, fasting blood glucose, triglyceride and obesity.

The most frequent MetS parameters were central obesity (40.44%); elevated TGs (40.19%) and hyperglycaemia (29.91%) followed by hypertension (29.65%) and decreased HDL-C (15.93% (Fig 1).

**Table 2. Behavioural risk factors among adult patients in Dessie comprehensive specialized hospital outpatient departments Dessie, Ethiopia 2020.**

| Variables | | Frequency (%) | MetS | |
|---|---|---|---|---|
| | | | Yes (%) | No (%) |
| Family history of CVDs | | | | |
| Yes | | 72(16.65) | 33(45.83) | 39(54.17) |
| No | | 336(83.35) | 110(32.7) | 226(67.3) |
| Current Smoker | Yes | 48 (11.8) | 17 (35.4) | 31 (64.6) |
| | No | 360 (88.2) | 126 (35) | 234 (65) |
| Ever Smoker | Yes | 58(14.2) | 22(37.9) | 36(62.1) |
| | No | 350(85.8) | 121(34.6) | 229(65.4) |
| Frequency of smoking | Daily | 37 (9.1) | 13 (35.1) | 24 (64.9) |
| | Occasionally | 11 (2.7) | 4 (36.4) | 7 (63.6) |
| | Non smoker | 360 (88.2) | 126 (35.0) | 234 (65.0) |
| Current alcohol user | Yes | 54 (13.2) | 25 (46.3) | 29 (53.7) |
| | No | 354 (86.8) | 118 (33.3) | 236 (66.7) |
| Type of drinker | Heavy drinkers | 30 (7.4) | 14 (46.7) | 16 (53.3) |
| | Moderate drinkers | 21 (5.1) | 10 (47.6) | 11 (52.4) |
| | No drinkers | 357 (87.5) | 119 (33.3) | 238 (66.7) |
| Coffee Consumption | Yes | 208 (51.0) | 72 (34.6) | 136 (65.4) |
| | No | 200 (49.0) | 71 (35.5) | 129 (64.5) |
| Frequency of coffee consumption | non consumers | 200 (49.0) | 71 (35.5) | 129 (64.5) |
| | irregular consumer | 39 (9.6) | 16 (41) | 23 (59) |
| | exactly once a day | 69 (16.9) | 26 (37.7) | 43 (62.3) |
| | more than once a day | 100 (24.5) | 30 (30) | 70 (70) |
| Khat chewing | Yes | 54(13.2) | 17(31.5) | 37(68.5) |
| | No | 354(86.8) | 83(23.4) | 271(76.6) |
| Physical activity | Low physical activity | 262 (64.2) | 118 (45.0) | 144 (55.0) |
| | Moderate Physical Activity | 40 (9.8) | 11 (27.5) | 29 (72.5) |
| | High level Physical Activity | 106 (26.0) | 14 (13.2) | 92 (86.8) |
| Spend of leisure time | Reading, watching TV, or other sedentary activity | 272 (66.7) | 99 (36.4) | 173 (63.6) |
| | Walking, cycling | 97 (23.8) | 35 (36.1) | 62 (63.9) |
| | Participation in recreational sports | 39 (9.5) | 9 (23.1) | 30 (76.9) |
| Sleeping duration(in hours) | Less than six | 15(3.7) | 12(80) | 3(20) |
| | Six to seven | 53(13) | 24(45.3) | 29(54.7) |
| | Eight to nine | 266(65.2) | 84(31.6) | 182(68.4) |
| | Ten and above | 74(18.1) | 23(31.1) | 51(68.9) |

## Factors associated with MetS

In the adjusted multivariable logistic regression analysis, MetS among the study subjects was significantly associated with their socio-demographic features (sex, age, education, and place of residence), behavioural factors (current alcohol consumption, physical activity level, and sleeping duration) and current body mass index of the participants were found be independent predictors (Table 4).

As age increases the probability of having MetS increases. In the current study, the odds of MetS among aged participants (age ≥60 years) was 18 folds higher than younger individuals (age < 30 years) [OR = 18.23; 95% CI: (6.66, 49.84)]. The odds of MetS among adults with sleeping duration less than six hours per day was about five times higher than the odds of MetS in adults who had a sleeping duration often and more hours per day [OR: 4.62; 95% CI: (1.02, 20.98)].

**Table 3. Dietary risk factors among adult patients in Dessie comprehensive specialized hospital outpatient departments Dessie, Ethiopia June 2020.**

| Variables | | | Frequency (%) | MetS | |
|---|---|---|---|---|---|
| | | | | Yes (%) | No (%) |
| Regular Meal patterns | Breakfast and Dinner only | | 3(0.7) | 1(33.3) | 2 (66.7) |
| | Breakfast, lunch and dinner | | 315(77.2) | 106(33.7) | 209(66.3) |
| | Breakfast, lunch, Snack and dinner | | 90(21.1) | 36(40) | 54(60) |
| Meal plan | Yes | | 39(9.6) | 13(33.3) | 26(66.7) |
| | No | | 369(90.4) | 130(35.2) | 239(64.8) |
| Eating styles of participants | Erratic eater | | 277(67.9) | 95(65.7) | 182(34.3) |
| | Time constraint | | 131(32.1) | 48(36.6) | 83(63.4) |
| Servings of Fruit and/or vegetables per day | | Less than five | 361 (88.5) | 131 (36.3) | 230 (63.7) |
| | | Five and above | 47 (11.5) | 12 (25.5) | 35 (74.5) |
| Oil or fat most often used | | Mixed | 42 (10.3) | 15 (35.7) | 27 (64.3) |
| | | Palm oil | 122 (29.9) | 38 (31.1) | 84 (68.9) |
| | | Vegetable oil | 244 (59.8) | 90 (36.9) | 154 (63.1) |
| Sugar and sweet | | Daily | 165 (40.4) | 61 (37.0) | 104 (63.0) |
| | | Occasionally | 218 (53.5) | 71 (32.6) | 147 (67.4) |
| | | Don't take | 25 (6.1) | 11 (44.0) | 14 (56.0) |
| Egg | | Daily | 49 (12.0) | 14(28.6) | 35 (71.4) |
| | | Occasionally | 340 (83.3) | 125 (36.8) | 215 (63.2) |
| | | Don't take | 19 (4.7) | 4 (21.1) | 15 (78.9) |
| Red meat | | Daily | 48 (11.8) | 16 (33.3) | 32 (66.7) |
| | | Occasionally | 317 (77.7) | 109 (34.4) | 208 (65.6) |
| | | Don't take | 43 (10.5) | 18 (41.9) | 25 (58.1) |
| Consumption outside home (café, restaurant or hotel) | | ≥4 times per week | 42(11.8) | 19(45.2) | 23(54.8) |
| | | 2–3 times per week | 8(2.5) | 4(50) | 4(50) |
| | | 1time per week | 27(6.6) | 10(37) | 17(63) |
| | | Only at home | 331(81.1) | 110(33.2) | 221(66.8) |
| Sugar & Sweet food intake | | Daily | 165(40.4) | 62(37.6) | 103(62.4) |
| | | Occasionally | 208(50.9) | 65(32.1) | 143(67.9) |
| | | Don't Take | 35(9.7) | 16(44) | 19(56) |
| Fried food | | Daily | 44 (10.8) | 16 (36.4) | 28 (63.6) |
| | | Occasionally | 313 (76.7) | 107 (34.2) | 206 (65.8) |
| | | Don't take | 51 (12.5) | 20 (39.2) | 31 (60.8) |
| Adult BMI category | | Obese | 33(8.2) | 18(54.5) | 15(45.5) |
| | | Overweight | 83(20.3) | 45(54.2) | 38(45.8) |
| | | Normal | 249(61) | 64(25.7) | 185(74.3) |
| | | Thin | 43(10.5) | 16(37.2) | 27(62.8) |

A dose-response analysis showed MetS had almost linear relationships with the log Odds ratios of sleep duration (Fig 2A), physical activity (Fig 2B), BMI (Fig 2C) and educational level (Fig 2D).

## Discussion

To begin with the study's pertinent findings, more than one-third of adults attending outpatient departments of DCSH in Ethiopia had MetS. Women, older age groups, urban residents and individual with lesser sleep duration, higher BMI, physical inactivity, alcohol consumption and high educational level were disproportionately affected by MetS.

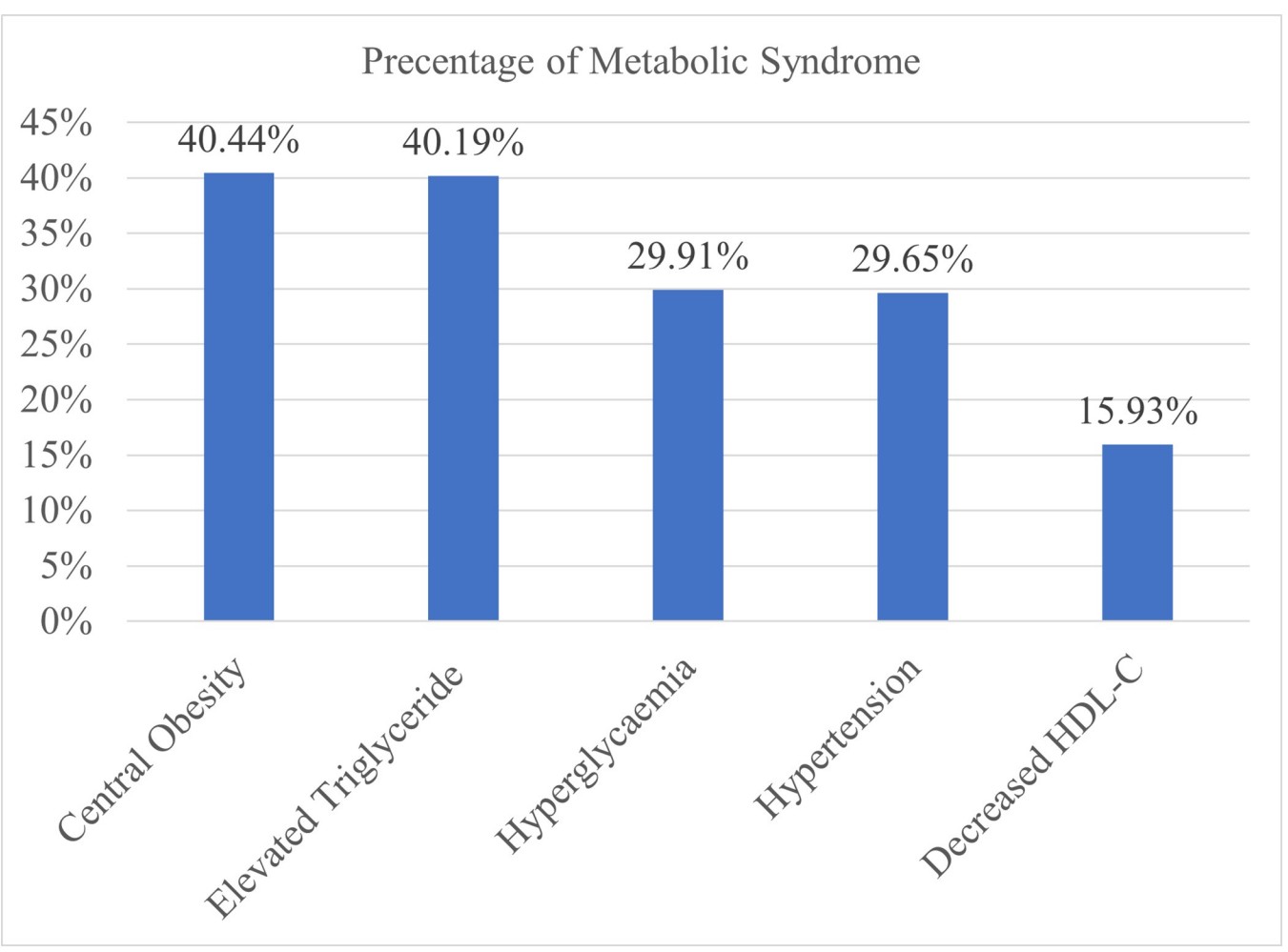

**Fig 1. Components of metabolic syndrome among adults in Dessie comprehensive specialized hospital outpatient departments Dessie, Ethiopia 2020.**

The percentage of adults having at least three of the five components of MetS was 35.0%. This finding is similar with a study conducted in Cameroon (32.45%) [34] but higher than studies conducted at St. Paul's Hospital, Addis Ababa (20.3%) [35], Jimma University Hospital, (26%) [14], Mizan-Aman town, Ethiopia (9.6%), Brazil (8.9%), Canada (19.1%) and Algeria (20%) [20, 36–38] and lower than studies done in University of Gondar hospital, Amhara (45.3%), Ayder Hospital, Tigray, Ethiopia (51.1%) and India (76%) [13, 16, 21]. These variations may be due to discrepancies in sample size; variations in existing interventions and available infrastructures, prolonged time gaps between studies, ethnic differences, and application of non-uniform tools to define MetS. Additionally, the possible reason for the higher prevalence of MetS in Brazil, Canada, and Algeria may be due to the difference in availability of effective nutrition policy and awareness level related to treatment and prevention of NCDs.

The current finding was supported by a review conducted in developing countries that revealed that rapid urbanization, nutrition transition, sedentary behaviour, and risky personal behaviours attributing for the highest burden of metabolic syndrome [12]. The odds of MetS among adults with low physical activity were four-fold higher compared with adults with high physical activity and in line with studies from Mizan Aman of the southern region, Addis Ababa and Tigray region of Ethiopia, rural northeast China, and Qatar [20, 21, 38–41]. This is

**Table 4. Factors associated with metabolic syndrome among adult patients in Dessie comprehensive specialized hospital outpatient departments Dessie, Ethiopia 2020.**

| Variables | | MetS | | COR(95%CI) | AOR(95%CI) |
|---|---|---|---|---|---|
| | | Yes(%) | No(%) | | |
| Sex | Female | 85(40.3) | 126(59.7) | 1.67(1.07,2.44) | 1.85(1.01,3.38)* |
| | Male | 58(29.4) | 139(70.6) | 1 | 1 |
| Age in years | 18–29 | 14(14.9) | 80(85.1) | 1 | 1 |
| | 30–39 | 16(20.0) | 64(80.0) | 1.43(0.75,2.60) | 2.26 (0.83, 6.13) |
| | 40–49 | 30(36.2) | 53(63.8) | 3.23(1.81,6.27) | 9.81 (3.73, 25.83)*** |
| | 50–59 | 44(60.3) | 29(39.7) | 2.26(1.08,4.70) | 39.67 (13.84, 113.6)*** |
| | ≥60 | 39(50.0) | 39(50.0) | 6.53(3.35,12.72) | 18.23(6.66, 49.84)*** |
| Educational status | No formal education | 14(20.9) | 53(79.1) | 0.36(0.14,0.73) | 0.30(0.12,0.74)** |
| | High school and less | 56(43.5) | 111(66.5) | 0.69(0.35,0.89) | 0.68(0.38,1.25) |
| | College and above | 73(42.0) | 101(58.0) | 1 | 1 |
| Resident | Urban | 110(39.7) | 167(60.3) | 1.96(1.23,3.11) | 1.94(1.08,3.24)* |
| | Rural | 33(25.2) | 98(74.8) | 1 | 1 |
| Physical activity | Low | 118(45.0) | 144(55.0) | 5.38(2.92,9.94) | 4.05 (1.80, 9.11)** |
| | Moderate | 11(27.5) | 29(72.5) | 2.49(1.02,6.08) | 2.59 (0.84, 8.02) |
| | High | 14(18.6) | 92(81.4) | 1 | 1 |
| BMI | Obese | 18(54.5) | 15(45.5) | 3.47(1.65,7.28) | 3.14 (1.20, 8.18)* |
| | Overweight | 45(54.2) | 38(45.8) | 3.42(2.04,5.74) | 2.04 (1.05,3.95)* |
| | Underweight | 16(37.3) | 27(62.7) | 1.71(.87,3.38) | 2.30 (0.95, 5.59) |
| | Normal | 64(25.8) | 185(74.2) | 1 | 1 |
| Current alcohol drunker | Yes | 25(46.3) | 29(53.7) | 1.72(1.02, 3.08) | 2.85(1.27,6.39)* |
| | No | 118(33.3) | 236(66.7) | 1 | 1 |
| Consumption of fruits and vegetables per week | Less than five times | 131(36.3) | 230(63.7) | 1.66(0.83, 3.31) | 2.28(0.94, 5.56) |
| | More than five times | 12(25.5) | 35(74.5) | 1 | 1 |
| Sleeping duration (in hours) | Less than six | 12(80.0) | 3(20.0) | 8.87(2.28,34.47) | 4.62(1.02, 20.98)* |
| | Six to nine | 108(33.9) | 211(66.1) | 1.83(0.88,3.81) | 1.26(0.53,2.98) |
| | Ten and above | 23(31.1) | 51(68.9) | 1 | 1 |
| | | | | | |

NB

*, ** and *** indicate at P-values at <0.05, <0.001 and <0.0001 respectively. The table was adjusted for marital status, family history, smoking, chewing khat, sugar and sweetened food intake, fried food consumption, meal plan and eating style

underpins that individuals without regular physical activity are at higher risk of elevated BP, insulin resistance, diabetes, dyslipidaemia, and obesity due to altered or reduced energy consumption, or positive energy balance [9, 45, 46].

Older adults had a higher probability of getting MetS and age is an independent risk factor for developing hypertension, but the changes in BP associated with aging are more pronounced in women compared to men. As age increases, the percentage of body fat also increases due to a change in body composition [21, 34, 36, 40, 42, 43, 47, 48]. The current study indicated that women were more affected by MetS than men and it was consistent with studies conducted in the United States and Portugal [48, 49] but in contrast to the above findings, some studies revealed that men were commonly presented with MetS [50–52]. This may be due to the presence of distinct differences in the prevalence of dysglycemia, body fat distribution, adipocyte size and function, hormonal regulation of body weight and adiposity, and the influence of oestrogen decline on risk factor clustering [51] and in general, data on differences in metabolic syndrome in men and women is scarce [47].

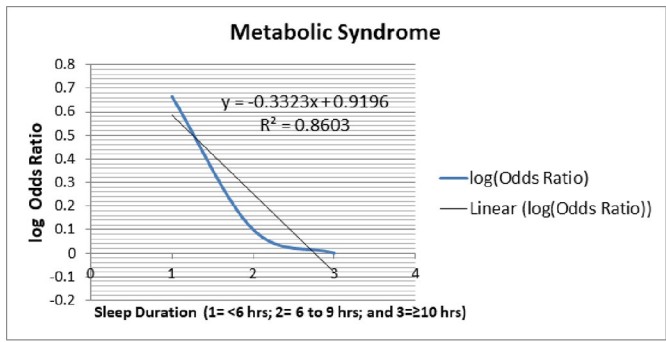 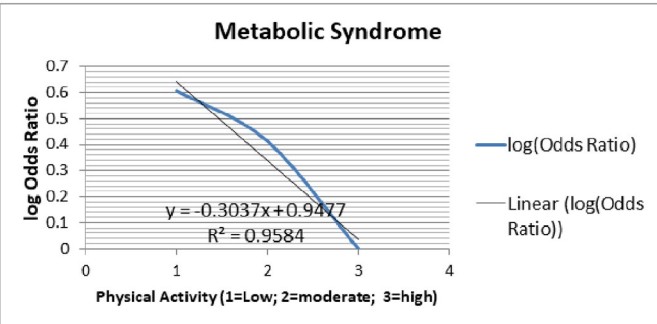

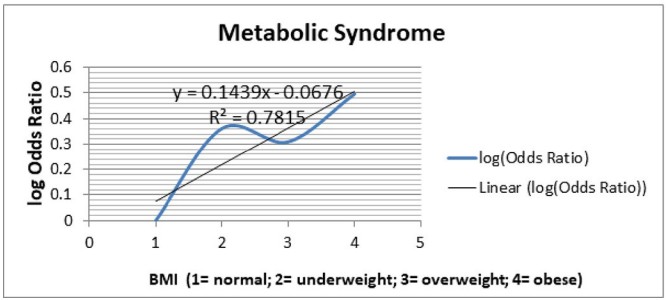 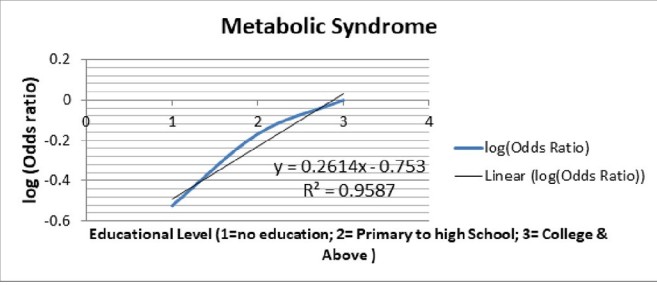

**Fig 2.** Nonlinear dose-response relationships of metabolic syndrome with a) sleeping duration, b) physical activity, c) BMI, and d) educational level among adults in Dessie Comprehensive Specialized Hospital, South Wollo, Ethiopia 2020.

The current study also identified that no or low educational status reduces the likelihood of having MetS which contradicts to other studies [48, 49, 53]. In the context of the current study, higher clustering of uneducated individuals are found in the rural areas who may not be affected by sedentary behaviour, consumption of energy-dense nutrients, and processed food commodities and may be due to the socio-economic inequalities

Furthermore, the current study reaffirmed that obese and overweight individuals were at greater risk of developing MetS. Whatever the study settings, this result was in line with other studies in Ethiopia, Cameroon, South India, Brazil, and Canada [21, 34, 36, 37, 44]. Obesity strongly linked to the alteration of the five diagnostic criteria and plays a crucial role in the development of MetS [9, 14]. Another risk personal behaviour that contributes to the occurrence of MetS in this study was habitual alcohol consumption. The finding was similar to findings conducted in Brazil, Venezuela, and China which showed that moderate to heavy drinking of alcohol leads to MetS [54–56] but other studies from the United States and China revealed that mild to moderate alcohol consumption reduces the risk of MetS [57, 58]. This may be due to the fact that alcohol is a concentrated source of energy and can distort the total energy pool of adults and it may also trigger individuals for aggressive eating conditions following drinking.

Additionally, the current study called for intervention to avoid insufficient sleep duration that leads to MetS. The study identified that adults with short sleeping duration (<6 hours/day) were independently associated with MetS which was congruent with many epidemiological studies and Systematic reviews [59, 60] even if the presence of paradoxical reports exist [61]. However, the underlying mechanisms between shorter sleep duration and MetS are not clear or well understood [60]. The potential biological mechanisms how change in circadian rhythm related to the development of MetS and NCDs need further investigation using strong methodological approach. Further experiments should be designed to validate and recommend optimal cut-off for sleep duration for the prevention of NCDs.

This study has limitations in assuring the accuracy of some information which are self-reported such as sleeping duration, alcohol consumption, physical activity and dietary habits. Even though detailed questionnaire was used to commemorate their past activities, recall bias is inevitable. The study also didn't address interrelationships or interaction effects of risky personal behaviour one over the other using causal pathways analysis. The use of standardized tools and laboratory procedures helps in producing reliable estimates and generalizing the findings to the community. Additionally, the referral hospital has been serving the wide catchment areas from Eastern Amhara and Afar regions and the use of similar management protocol for MetS across the nation, the findings of study can be inferred to populations attending OPDs in Ethiopia. Moreover, a more mechanistic longitudinal study is deemed necessary to confirm the relationships of sleep duration and metabolic syndrome to reach definitive conclusion.

## Conclusions

This study revealed a growing epidemic burden of MetS in Ethiopia and has become one of the major health challenges worldwide. The substantial gender difference was noted that the overall MetS was almost two folds higher in women than men. The common component of MetS was central obesity followed by elevated triglycerides. In general sleep duration, physical activity, BMI and educational level of participants have linear relationships with MetS. MetS in the general adult population was highly contributed due to rapid urbanization, demographic transition, personal behavioural factors, and nutrition. In the current study personal lifestyle or behavioural factors predominantly contributed to the rapid increment of MetS and should be given due attention for large-scale interventions. The national NCD prevention strategy should be reframed in addressing the modifiable risk factors for such cardiometabolic disease to minimize and avert morbidity and mortality burden at population level.

## Acknowledgments

We acknowledge data collectors, supervisors and study participants for their trusted and cooperative response to successful accomplishment of the field work.

## Author Contributions

**Conceptualization:** Mulugeta Belayneh, Tefera Chane Mekonnen, Sisay Eshete Tadesse, Erkihun Tadesse Amsalu, Fentaw Tadese.

**Data curation:** Mulugeta Belayneh, Tefera Chane Mekonnen, Erkihun Tadesse Amsalu, Fentaw Tadese.

**Formal analysis:** Mulugeta Belayneh, Tefera Chane Mekonnen, Sisay Eshete Tadesse.

**Funding acquisition:** Mulugeta Belayneh, Tefera Chane Mekonnen, Sisay Eshete Tadesse, Erkihun Tadesse Amsalu.

**Investigation:** Mulugeta Belayneh, Tefera Chane Mekonnen, Sisay Eshete Tadesse, Erkihun Tadesse Amsalu.

**Methodology:** Mulugeta Belayneh, Tefera Chane Mekonnen, Sisay Eshete Tadesse, Fentaw Tadese.

**Project administration:** Mulugeta Belayneh, Tefera Chane Mekonnen, Erkihun Tadesse Amsalu.

**Resources:** Mulugeta Belayneh, Tefera Chane Mekonnen, Sisay Eshete Tadesse, Erkihun Tadesse Amsalu, Fentaw Tadese.

**Software:** Mulugeta Belayneh, Tefera Chane Mekonnen, Fentaw Tadese.

**Supervision:** Mulugeta Belayneh, Tefera Chane Mekonnen, Sisay Eshete Tadesse.

**Validation:** Mulugeta Belayneh, Tefera Chane Mekonnen, Fentaw Tadese.

**Visualization:** Mulugeta Belayneh, Tefera Chane Mekonnen, Fentaw Tadese.

**Writing – original draft:** Mulugeta Belayneh, Tefera Chane Mekonnen, Sisay Eshete Tadesse, Erkihun Tadesse Amsalu.

**Writing – review & editing:** Mulugeta Belayneh, Tefera Chane Mekonnen, Sisay Eshete Tadesse, Erkihun Tadesse Amsalu, Fentaw Tadese.

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
