## [Decision Letter · Decision Letter 0]

4 Apr 2022

PONE-D-21-20970Sleeping Duration, Physical Activity and Alcohol Drinking as Potential Attributes of Metabolic Syndrome in Adults in EthiopiaPLOS ONE

Dear Dr. Mekonnen,

Thank you for submitting your manuscript to PLOS ONE. After careful consideration, we feel that it has merit but does not fully meet PLOS ONE’s publication criteria as it currently stands. Therefore, we invite you to submit a revised version of the manuscript that addresses the points raised during the review process.

 Five reviewers have reviewed your paper and made several excellent suggestions for improvement. Please pay specific attention to:Use of the central obesity measure and cut-offs, clarify, and add additional analyses using different cut-offs.Identification of U-shaped relationships should be based on the multivariable analysis. Looking at the adjusted ORs no U-shape relationships were identified though. If there are any U-shape relationships, please present them graphically using log odds or log odds ratio plots for dose - response non-linear continuous exposure.Better describe choice and definition of risk factors, as well provide more details about the study sites.Make sure to adhere to STROBE reporting guidelines for cross-section studies.Also note that reviewer 2 included the review of your paper as a separate Word document.Please submit your revised manuscript by May 19 2022 11:59PM. If you will need more time than this to complete your revisions, please reply to this message or contact the journal office at plosone@plos.org. Please include the following items when submitting your revised manuscript:A rebuttal letter that responds to each point raised by the academic editor and reviewer(s). You should upload this letter as a separate file labeled 'Response to Reviewers'.A marked-up copy of your manuscript that highlights changes made to the original version. You should upload this as a separate file labeled 'Revised Manuscript with Track Changes'.An unmarked version of your revised paper without tracked changes. You should upload this as a separate file labeled 'Manuscript'.

We look forward to receiving your revised manuscript.

Kind regards,

Bart Ferket

Academic Editor

PLOS ONE

Journal Requirements:

2. Thank you for stating the following financial disclosure: "NO- The funders had no role in study design, data collection and analysis, decision to publish, or preparation of the manuscript."

3. Thank you for stating the following in the Acknowledgments Section of your manuscript: "First, we would like to thank Wollo University for giving opportunity and over all support to do this research.  I also acknowledge Dessie comprehensive specialized Hospital for covering of full laboratory cost of the respondents and any necessary support to conduct this research. I also appreciate my data collectors and my study participants for their trusted and cooperative response to my questionnaires. The last but not the least I would like to thank all individuals; my beloved family, and Dessie comprehensive specialized hospital staffs who helped me."

Please remove any funding-related text from the manuscript and let us know how you would like to update your Funding Statement. Currently, your Funding Statement reads as follows: "NO- The funders had no role in study design, data collection and analysis, decision to publish, or preparation of the manuscript."

Reviewers' comments:

Reviewer's Responses to Questions

**Comments to the Author**

1. Is the manuscript technically sound, and do the data support the conclusions?

Reviewer #1: Yes

Reviewer #2: Yes

Reviewer #3: Partly

Reviewer #4: Partly

Reviewer #5: Yes

2. Has the statistical analysis been performed appropriately and rigorously? 

Reviewer #1: Yes

Reviewer #2: I Don't Know

Reviewer #3: I Don't Know

Reviewer #4: Yes

Reviewer #5: Yes

3. Have the authors made all data underlying the findings in their manuscript fully available?

Reviewer #1: Yes

Reviewer #2: Yes

Reviewer #3: Yes

Reviewer #4: No

Reviewer #5: Yes

4. Is the manuscript presented in an intelligible fashion and written in standard English?

Reviewer #1: No

Reviewer #2: Yes

Reviewer #3: No

Reviewer #4: Yes

Reviewer #5: Yes

5. Review Comments to the Author

Reviewer #1: 1. The title: should be reconsidered. Looking at the line 235 of the manuscript, several factors are associated with the Metabolic syndrome .eg sex and age, yet the author chooses only a few for the title. They may consider "associated factors" without being specific.

2. Abstract: The conclusion on the abstract would need revision. in Line 44 the author uses the words " very emerging " which does not sound right given that we are not comparing it with any specific time in the past , to provide evidence of an increase.

3. Introduction: Would be useful to justify in a paragraph why the author chose a hospital population and not a community sample.

4. Methods: Line 101 need to be checked. The author says " simple random sample to catch up on the subjects" which does not sound very scientific.

Line 105: Why did the author choose WHO Steps survey, would help to describe this a bit more and provide reference.

Line 125 : What is " Ethiopia's adult classification System"

Define first terms the first time they appear, See FBD, HDL ( lines 131, 135).

Line 141 Give reference for General physical Questionnaire and report if it has been used previously in similar settings.

Line 156: Give reference for the food questionnaire

163: Justify the fact that the one question on sleep is adequate.

5. Discussion: include limitations of the study.

Reviewer #2: I have offered specific comments, please find attached review report. Thus, improving the clarity of the study’s reporting guidelines will, in my mind, undoubtedly increase its impact. The study design methodology is currently missing crucial information that should be pre-specified comprehensively and as far as possible in the abstract and methods section.

Reviewer #3: 1. Abstract: line 45 - " U " shaped relationships. Relationship for e.g between physical activity and MetS is not U shaped. Low, moderate and high physical activity - The prevalence of Mets is 45%, 27.5% and 13.2 % respectively. Similarly with sleep duration and education level

2. Inconsistent use of waist circumference cut -off values . Line 63-64 - 83.7 cm and 78 cm for males and females respectively - Authors use country specific WC cut off. However, Line 126- 127 , - Authors use IDF criteria 94cm and 80 cm respectively for males and females

3. Sleep categories inconsistent and incorrectly described . Lines 164 -165 - 4 categories noted ,However only 3 are described. Furthermore, the categories in Table 2 are not consistent with those described in the method (lines 164 -165 ), or in the publication that the authors use as a reference

4. Line 169 - Variables with a p value of less than 0.2 in the bivariate analysis were exported to the final model. Although I am aware that it is not incorrect to use this level of significance , can the authors please explain the rationale behind the use of this level of significance in this study.

5. Line 256 -this statement is not correct in two respects.

a. The results do not support this statement. The authors state that there was an inverse relationship between level of education and MetS. However, in Table 2 the prevalence of MetS is 20.1 % ,33.5% and 41.9 % in patients with no education, a high school level of education and a college education respectively

b. The publication referenced does not contradict this statement. Santos et al (reference 48) indeed reported an inverse relationship between educational level and the prevalence of MetS in their study

6. Table 1 -No currency for monthly income level categories

7. Table 2 - Poorly formatted.

8. Table 3 - "Consumption outside home " needs to be clarified.

9. Table 5 - The prevalence of MetS should be presented as n(%) ,not only n

10. An examples of a significant language error

lines 287- 289 : "However, the clear underlying mechanisms related to the relationship between shorter sleep duration and MetS are understood."

I believe the authors meant to say the underlying mechanisms between duration of sleep and MetS are not clear or well understood

Reviewer #4: This is a very important topic area, metabolic syndrome and its associated risk factors are required in SSA population. However, authors should consider the following suggestions.

1. I think the current title over exaggerates the study, because other risk factors of metabolic syndrome were looked at in the study, why is the author focusing on sleep duration, alcohol drinking and physical activity in the title.

2. Line 206, the author referenced table 4 twice, Table Four ….., S, and BP (Table 4). Please reword

3. The central obesity parameter that was used in this manuscript for African population might have over estimated or underestimated the prevalence among your population. The author have used .. waist circumference ≥83.7 64 cm for males and ≥78.0 cm for females. The harmonized definition suggested that 94 and 80cm should be used for SSA population (https://www.ahajournals.org/doi/10.1161/CIRCULATIONAHA.109.192644?url_ver=Z39.88-2003&rfr_id=ori:rid:crossref.org&rfr_dat=cr_pub%20%200pubmed) . Also an optimal waist circumference cut-off for SSA population has been published by Ekoru et al, 2018(https://www.nature.com/articles/ijo2017240). I will suggest that the authors should re-analyse using either of this cut off.

4. Line 214 .. Aas one becomes aged the probability of having MetS increased as well. Please reword this sentence

5. Generally, I will suggest that the Author edits the manuscript or engage an editor to complete that role on their behalf.

6. Line 287 – 289 .. It seems that both the genetic and environmental factors causing MetS but 288 in this study, the environmental factors predominantly contributed to the rapid increment of 289 MetS and are fully subjected to modification given large-scale interventions are designed. I will suggest that the author should conclude the study following the focus of the variables that were collected, which in this instance are behavioural factors. The author has not collected r discussed any environmental factors for this study, so should not conclude

Reviewer #5: The metabolic syndrome is an increased public health issue particularly in low- and middle-income countries (as authors noted). It needs to be well described in both general population and specific population (like patients). The study is of important for both clinician and public health managers in Ethiopia to address MetS in all context.

Major comments

• Authors have noted (line 83-84) that “Previous studies conducted in Ethiopia have documented a high prevalence of MetS ». Thus, it is not clear why the study was conducted and which gap of knowledge it filled.

• Line 94-95: Study site was not described. Authors have to provide more details about the study site (is it a secondary of tertiary hospital? how many bed?, which department of hospital was targeted and why? How many outpatients attended by year? the hospital is located in rural or urban areas, etc.…)

• Line 99: it is not clear how the sample size was calculated and how the simple ramdom sampling was performed how the study design was considered in the data analysis. Can authors provide more information about this?

• I suggest that authors improve the presentation of the results of their study, mainly subsection about the MetS (line 198-207). Indeed, they have to describe the prevalence of MetS in a paragraph and the components of MetS in another paragraph. In addition, It is not clear why authors check for a correlation between WC and others MetS components since this was not cited as study objective. I recommend the authors to be focus on their study objective in the presentation of the results of their study.

• Authors have to provide limitations of their study since the study has many limitations. First, it was conducted in hospital and may not be considered to make inference to the general population of Ethiopia. The behavioral variables might be affected by social desirability bias.

• Authors has to provide more details of public health or clinical pratice implication of their study since it was done in health facility. e.g: to organize sensitization campaign targeting patients attended in hospital.

Minors comments

• I suggest that authors in the title of the manuscript precise the type of the study

• Line 56: Please, correct the abbreviation of metabolic syndrome “MetS” instead of “Mets”

• Line 63: Please, precise which definition of metabolic syndrome was provided here. It is not clear in the text.

• Line 68-69: authors had to indicate why they used the most recent definition (since it is also discussed)

• line 73 to 75: Please, provide the reference of the statement

• Line 133: the mean of (26) is not clear.

• Line 172: the acronym AOR was use for the first, it needs to be defined.

• Line 186-190: It is not clear if the prevalence of MetS reported in this paragraph is the history of MetS or MetS diagnosis during the current study. Please, can authors provide clarification?

• Line 214: correct Aas

• Table 3 line 1 : Please the number 3(0.7)

• Please improve the tables formatting

6. PLOS authors have the option to publish the peer review history of their article (what does this mean?). If published, this will include your full peer review and any attached files.

Reviewer #1: No

Reviewer #2: **Yes: **Sphamandla Josias Nkambule

Reviewer #3: No

Reviewer #4: No

Reviewer #5: **Yes: **Kadari Cisse

---

## [Author Response · Author response to Decision Letter 0]

1 Jun 2022

We have attached a point-by-point responses for reviewers.

---

## [Decision Letter · Decision Letter 1]

7 Jul 2022

PONE-D-21-20970R1Sleeping Duration, Physical Activity and Alcohol Drinking as Potential Attributes of Metabolic Syndrome in Adults in Ethiopia:  A Hospital-Based Cross-Sectional Study.PLOS ONE

Dear Dr. Mekonnen,

Thank you for submitting your manuscript to PLOS ONE. After careful consideration, we feel that it has merit but does not fully meet PLOS ONE’s publication criteria as it currently stands. Therefore, we invite you to submit a revised version of the manuscript that addresses the points raised during the review process.Please revise the title as suggested by Reviewer 1, e.g., into: "Sleeping Duration, Physical Activity, Alcohol Drinking and Other Risk Factors as Potential Attributes of Metabolic Syndrome in Adults in Ethiopia: A Hospital-Based Cross-Sectional Study."Please revise the limitations and conclusions as suggested by Reviewer 1 as well.

We look forward to receiving your revised manuscript.

Kind regards,

Bart Ferket

Academic Editor

PLOS ONE

Journal Requirements:

Reviewers' comments:

Reviewer's Responses to Questions

**Comments to the Author**

1. If the authors have adequately addressed your comments raised in a previous round of review and you feel that this manuscript is now acceptable for publication, you may indicate that here to bypass the “Comments to the Author” section, enter your conflict of interest statement in the “Confidential to Editor” section, and submit your "Accept" recommendation.

Reviewer #1: All comments have been addressed

Reviewer #2: All comments have been addressed

2. Is the manuscript technically sound, and do the data support the conclusions?

Reviewer #1: Yes

Reviewer #2: Yes

3. Has the statistical analysis been performed appropriately and rigorously? 

Reviewer #1: Yes

Reviewer #2: Yes

4. Have the authors made all data underlying the findings in their manuscript fully available?

Reviewer #1: Yes

Reviewer #2: Yes

5. Is the manuscript presented in an intelligible fashion and written in standard English?

Reviewer #1: No

Reviewer #2: Yes

6. Review Comments to the Author

Reviewer #1: Additional METS STUDY COMMENTS

1. I still feel the Title needs to be reflective of the content of the manuscript. Because the authors accepts that fact, they should coin a title that includes more than sleep, physical activity and alcohol.

2. Given that only one question was asked about sleep, I propose that this is captured as part of the limitations.

3. I appreciate the inclusion of the limitation section-the statement “ undoubtedly reproducible” on line 345 can be replace with more gentle statement like“ may be generalized..because….”

4. The last two sentences on the conclusion section do not sound appropriate They can be reworded to encourage not sound like a warning “Unless the national health policy for NCDs prevention is revised and effectively implemented, the developmental progress of the nation will be stepped backed due to the high health care expenditure, high disability adjusted life years and economic crisis. In general the global commitment designed targets in the Sustainable Development Goals will be off-track by 2030”

Reviewer #2: Thank you, for affording me the opportunity to review your work. This manuscript is a herculean effort and an enjoyable read. The study’s objectives are essential in understanding the prevalence of metabolic syndrome and patterns of attributable risk factors in adults.

Moreover, this topic is significant, considering the more ageing adult population’s unprecedented growth and the projected increase in the prevalence of metabolic syndrome globally. After rigorously reviewing the revised manuscript, the authors have adequately addressed my comments raised in a previous round of review and you feel that this manuscript is now acceptable for publication.

Kudos, to the team!

7. PLOS authors have the option to publish the peer review history of their article (what does this mean?). If published, this will include your full peer review and any attached files.

Reviewer #1: No

Reviewer #2: **Yes: **Sphamandla Josias Nkambule

---

## [Author Response · Author response to Decision Letter 1]

7 Jul 2022

We have addressed the comments given by the reviewers and attached a file.

---

## [Editor Report · Decision Letter 2]

12 Jul 2022

Sleeping Duration, Physical Activity, Alcohol Drinking and Other Risk Factors as Potential Attributes of Metabolic Syndrome in Adults in Ethiopia:  A Hospital-Based Cross-Sectional Study

PONE-D-21-20970R2

Dear Dr. Mekonnen,

We’re pleased to inform you that your manuscript has been judged scientifically suitable for publication and will be formally accepted for publication once it meets all outstanding technical requirements.

Kind regards,

Bart Ferket

Academic Editor

PLOS ONE
---

## [Editor Report · Acceptance letter]

2 Aug 2022

PONE-D-21-20970R2 

Sleeping Duration, Physical Activity, Alcohol Drinking and Other Risk Factors as Potential Attributes of Metabolic Syndrome in Adults in Ethiopia:  A Hospital-Based Cross-Sectional Study 

Dear Dr. Mekonnen:

I'm pleased to inform you that your manuscript has been deemed suitable for publication in PLOS ONE. Congratulations! Your manuscript is now with our production department. 

Kind regards, 

on behalf of

Dr. Bart Ferket 

Academic Editor

PLOS ONE